# Synthesis and Characterization of PCL-Idebenone Nanoparticles for Potential Nose-to-Brain Delivery

**DOI:** 10.3390/biomedicines11051491

**Published:** 2023-05-22

**Authors:** Radka Boyuklieva, Asya Hristozova, Bissera Pilicheva

**Affiliations:** 1Department of Pharmaceutical Sciences, Faculty of Pharmacy, Medical University of Plovdiv, 4002 Plovdiv, Bulgaria; radka.boyuklieva@phd.mu-plovdiv.bg (R.B.); asya.hristozova@mu-plovdiv.bg (A.H.); 2Research Institute, Medical University of Plovdiv, 4002 Plovdiv, Bulgaria; 3Department of Analytical Chemistry and Computational Chemistry, Faculty of Chemistry, University of Plovdiv “Paisii Hilendarski”, 4000 Plovdiv, Bulgaria

**Keywords:** idebenone, poly-ε-caprolactone, nanoparticles, nose-to-brain delivery

## Abstract

The present work is focused on the preparation of an optimal model of poly-ε-caprolactone nanoparticles as potential carriers for nasal administration of idebenone. A solvent/evaporation technique was used for nanoparticle preparation. Poly-ε-caprolactone with different molecular weights (14,000 and 80,000 g/mol) was used. Polysorbate 20 and Poloxamer 407, alone and in combination, were used as emulsifiers at different concentrations to obtain a stable formulation. The nanoparticles were characterized using dynamic light scattering, SEM, TEM, and FTIR. The resulting structures were spherical in shape and their size distribution depended on the type of emulsifier. The average particle size ranged from 188 to 628 nm. The effect of molecular weight and type of emulsifier was established. Optimal models of appropriate size for nasal administration were selected for inclusion of idebenone. Three models of idebenone-loaded nanoparticles were developed and the effect of molecular weight on the encapsulation efficiency was investigated. Increased encapsulation efficiency was found when poly-ε-caprolactone with lower molecular weight was used. The molecular weight also affected the drug release from the nanostructures. Dissolution study data were fitted into various kinetic models and the Korsmeyer–Peppas model was found to be indicative of the release mechanism of idebenone.

## 1. Introduction

Neurodegenerative disorders (NDs) are characterized by progressive damage in neurons and neuronal loss, leading to motor dysfunction and/or cognitive deficiency. Oxidative stress, chronic inflammation, and mitochondrial dysfunctions seem to be frequently involved in NDs, mainly in Alzheimer’s disease (AD) and Parkinson’s disease (PD) [1,2,3,4]. Oxidative stress is a phenomenon resulting from an imbalance between the production and accumulation of reactive oxygen species (ROS) in cells and tissues and the ability of cellular mechanisms to eliminate these byproducts. The main consequences of oxidative stress in these diseases are loss of glutathione, oxidative damage to DNA, proteins, and lipids. Tissues with high rates of metabolism such as the brain, heart, liver, and kidneys require high levels of antioxidants. Numerous studies have reported beneficial effects of antioxidant therapy in AD [5,6] and PD [7]. Oxidative stress in the brain coupled with weakened antioxidant defenses can significantly impair functions of cellular organelles such as mitochondria, which results in the loss of synapses and cell death and leads to manifestation of the clinical symptoms associated with NDs [8].

Idebenone (IDB, Figure 1a), a short-chain benzoquinone, is a coenzyme Q10 analogue with a potent antioxidant activity. In cells and tissues, IDB is reduced by NAD(P)H quinone oxidoreductase 1 (NQO1) to the stable hydroquinone form, which is regarded as the active form of the molecule [9]. When activated, the IDB molecule donates electrons to detoxify radicals and to support ATP synthesis in the mitochondrial respiratory chain [10]. Like all quinones, it protects cell membranes and mitochondria from the action of ROS, prevents lipid peroxidation, and minimizes oxidative stress processes. Several articles have provided completely new insights into the mode of action of IDB. IDB has been reported to reduce inflammation in various test systems, such as neuroinflammation [11,12], nanotoxicity [13], atherosclerosis [14], and drug-induced inflammation [15]. These effects cannot be explained by antioxidant activity or an increase in ATP synthesis alone. Inflammation is usually associated with hypoxia. It was recently described that, under hypoxia-reperfusion conditions, IDB prevents mtDNA release and subsequent activation of the inflammasome LNRP3, allowing IDB to intervene in one of the earliest steps of the pro-inflammatory cascade [12]. Inflammation also has mitochondrial involvement, suggesting that inflammatory conditions of the CNS may be therapeutically targeted with IDB. Based on a new understanding of the molecular actions of this drug, it can be expected that new indications for IDB will be identified. 

IDB was originally developed to treat AD and other cognitive impairments. Clinical trials show a lack of proven efficacy [16]. After oral administration, IDB is rapidly absorbed from the gut and extensively metabolized in the liver, so that less than 1% reaches the circulation. This rate can be improved by incorporating IDB into various liquid formulations, such as suspensions, and by administering it together with fatty food [17]. Even when taking these measures, bioavailability is less than 14% in humans. More than 99% of the circulating drug is bound to plasma proteins, which also reduces brain accumulation and hinders therapeutic potential. 

IDB has a molecular weight of 338.44 g/mol, an aqueous solubility of 0.0163 mg/mL, and log P of 3.57 (DRUGBANK^®^). Alternative dosage forms have been developed to improve the oral bioavailability of IDB, such as liposomes [18], complexes with cyclodextrins [19], and nanorods [20]. All formulations showed increased water solubility of the drug and improved oral absorption. However, other strategies can be used to increase the distribution of IDE in the brain, making the treatment of CNS disorders more effective with respect to conventional forms. For example, intranasal administration of IDB-loaded polymeric nanoparticles (NPs) can provide direct delivery of drug molecules to the brain tissue via the olfactory and trigeminal nerve pathways, bypassing the blood–brain barrier (BBB) and the pre-systemic gastrointestinal and hepatic elimination [21]. In addition, nose-to-brain drug transport is known to be very efficient in humans [22]. In this way, the treatment of specific neurodegenerative disorders such as AD and PD may be more successful.

The mechanism of drug transport from the nose to the brain via the olfactory pathway is still unclear. Scientists have not investigated whether the drug is released in the nasal cavity and transported to the brain or whether the drug carriers are delivered directly via a neural connection [23]. Most of the research has focused on increasing drug concentration in the brain and improving efficacy rather than evaluating drug transport mechanisms [24]. The transport of formulations depends on their size, morphology, and surface characteristics [21]. 

Poly-ε-caprolactone (PCL, Figure 1b) is a biodegradable polyester with a low melting point of around 60 °C. It is usually prepared by ring-opening polymerization of ε-caprolactone. PCL degrades by hydrolysis of its ester linkages under physiological conditions and has therefore received much attention for use as an implantable biomaterial. Approved by the US Food and Drug Administration (FDA), PCL is used to develop drug delivery devices, sutures, or adhesion barriers [25].

The aim of the present work was based on the design, development, and physicochemical characterization of biodegradable IDB-loaded PCL NPs, which can provide direct delivery of IDB to the CNS via intranasal administration. Additionally, the effect of the process variables emulsifier type and polymer molecular weight on structure, morphology, and entrapment efficiency was investigated.

## 2. Materials and Methods

### 2.1. Materials

Idebenone (IDB, Mw 338.44 g/mol), polymer poly-ε-caprolactone (PCL, Mw 14,000, 80,000 g/mol), Poloxamer 407 (Mw 12,600 g/mol), Polysorbate 20 (Mw 1220 g/mol) were purchased from Sigma-Aldrich (St. Louis, MO, USA). All other reagents and solvents were of analytical grade and used as provided.

### 2.2. Methods

#### 2.2.1. Preparation of NPs

PCL NPs were obtained by single emulsion/solvent evaporation technique [26]. For the preparation of the organic phase of the emulsion, PCL was dissolved in dichloromethane (DCM) followed by sonication in a bath sonicator (Sonorex Bandelin electronic, Berlin, Germany) until complete dissolution of the polymer. The oil-in-water (O/W) emulsion type was chosen as suitable for the inclusion of water-insoluble substances such as IDB. To form a single O/W emulsion, the polymer solution was added to the water phase containing the emulsifiers (Poloxamer 407 or/and Polysorbate 20) under high-speed homogenization at 25,000 rpm (Miccra MiniBatch D-9, MICCRA GmbH, Heitersheim, Germany) for 3 min. The obtained nanoemulsion was then stirred on an overhead stirrer (HS-100D, Witeg Labortechnik GmbH, Wertheim, Germany) with a 3-bladed propeller at 800 rpm until complete evaporation of DCM and solidification of the PCL droplets. Centrifugation was performed in Amicon^®^ tubes (Merck KGaA, Darmstadt, Germany) at 3800× *g* for 20 min at 20 ± 1 °C to attain nanoparticle pellets. To prepare drug-loaded NPs, IDB was dissolved into the organic phase prior to emulsification. The same procedure was then followed to collect the NPs.

#### 2.2.2. Particle Size Analysis, Size Distribution, and Zeta Potential 

The particle size of the obtained IDB NPs was determined by dynamic light scattering (DLS) method using a Nanotrac particle size analyzer (Microtrac, York, PA, USA). The analyzer is equipped with 3 mW helium/neon laser operating at 780 nm wavelength, which measures the particle size using noninvasive backscattering technology. Particle size analysis is carried out in the range of 0.8 nm to 6.5 µm. Samples from the nanoparticle suspension were directly analyzed. All the measurements were performed at 25.0 °C at 20 s intervals and were repeated three times.

#### 2.2.3. Scanning Electron Microscopy (SEM)

Visualization of the nanoparticles was carried out by scanning electron microscopy (Prisma E SEM, Thermo Scientific, Waltham, MA, USA). The samples were dried and then loaded on aluminium stubs and sputter coated with gold using a vacuum evaporator (Q150T ES Plus, Quorum Technologies, West Sussex, UK). The images were obtained at 15 kV acceleration voltage at 20,000× magnification using an Everhart–Thornley detector (ETD).

#### 2.2.4. Transmission Electron Microscopy (TEM) 

The morphology and size of PCL NPs were also observed by TEM. A drop of the nanosuspension was placed on a 200 mesh formvar-coated copper grid and allowed to dry for 24 h. Images were obtained using Talos F200C G2 Transmission Electron Microscope (Talos 1.15.3, Thermo Fisher Scientific, Waltham, MA, USA) operating at 200 kV and analyzed using Velox Imaging Software (Velox 2.15.0.45, Waltham, MA, USA).

#### 2.2.5. Fourier-Transform Infrared (FTIR) Spectroscopy

FTIR spectroscopy was used to investigate compatibility between the drug and the polymer. The spectra were collected using a Nicolet iS 10 FTIR spectrometer equipped with a diamond attenuated total reflection (ATR) accessory (Thermo Fisher Scientific, Pittsburgh, PA, USA). Analyses were performed in the range from 600 cm^−1^ to 4000 cm^−1^ with a resolution 4 nm and 32 scans. The spectra were analyzed with the OMNIC^®^ software package (Version 7.3, Thermo Electron Corporation, Madison, WI, USA).

#### 2.2.6. Estimation of IDB Loading (DL) and Entrapment Efficiency (EE)

IDB determination was carried out by high-performance liquid chromatography (HPLC). To evaluate the entrapment efficiency (EE%), a direct method based on ultrafiltration was used. The free IDB was separated from IDB-loaded NPs after centrifugation (Sigma Centrifuge 3–18 KS, Osterode am Harz, Germany) performed in Amicon^®^Ultra-15 tubes (100 K MWCO) (Merck KGaA, Darmstadt, Germany) for 20 min at 3800× *g* at 20 ± 1 °C. The separated NPs were suspended in ethanol and the mixture was sonicated for 15 min to allow extraction of IDB from the particles. The solutions were filtered through 0.45 μm PTFE syringe filters (Isolab Laborgeräte Gmb, Eschau, Germany) and analyzed. The amount of IDB was determined by HPLC (UltiMate 3000, Thermo Scientific, Waltham, MA, USA) using an Inertsil^®^ ODS-3HPLC column (150 × 4.6 mm, 5 µm, GL Sciences, Tokyo, Japan) under the following conditions: 281 nm wavelength, 10 μL injection volume, 1 mL/min flow rate, 25 °C oven temperature, mobile phase methanol:water in ratio 80:20. The retention time of IDB was 6.58 min, the total run time was 8 min. For comparative study, the ultrafiltrates were also analyzed. Aliquots of 1 mL of ultrafiltrate were diluted with ethanol, filtered, and chromatographed. The actual drug content was calculated from the calibration curve. The encapsulation efficiency was estimated using the following equation:(1)$$EE\left(\%\right)=\frac{Actual\;drug\;content}{Theoretical\;drug\;content}\times 100$$

The drug loading was calculated according to equation:(2)$$DL\left(\%\right)=\frac{Actual\;drug\;content}{Polymer\;amount+Actual\;drug\;content}\times 100$$

#### 2.2.7. In Vitro Drug Release Studies and Release Kinetics

In vitro release study was performed using the dialysis bag method. A dialysis membrane MWCO 12 kDa (Sigma-Aldrich Chemie GmbH, Taufkirchen, Germany) was cut into pieces (6 × 2.5 cm^2^) and hydrated in distilled water for 24 h. An accurately weighed amount of the nanosuspension (equivalent to 5 mg) was dispersed in 2 mL of PBS buffer (pH 7.4, corresponding to the physiological pH of the cerebrospinal fluid) and transferred into the dialysis bag, which was closed using a plastic clamp. Each bag was placed into a beaker containing 18 mL dissolution media (PBS buffer, pH 7.4) and kept on an electromagnetic stirrer at 100 rpm and 37 ± 0.5 °C. Samples of 1 mL were withdrawn at predetermined time intervals and replaced with equivalent volume of fresh media. The samples were diluted with 1 mL ethanol, then filtered through 0.45 μm PTFE syringe filters (Isolab Laborgeräte Gmb, Eschau, Germany) and analyzed for drug content as described in Section 2.2.6. Mean results of triplicate measurements and standard deviation were reported. 

## 3. Results and Discussion 

### 3.1. Synthesis and Characterization of Blank PCL NPs

The optimum conditions of the applied technique were derived after preliminary experimental work (Table 1). In that preliminary study, the oil/water ratio, the stirring rate, and the polymer and surfactant concentration were varied and their effect on the emulsion stability was investigated. Breakup of the emulsion and increase in droplet size were considered indicative of emulsion instability. PCL concentration of 0.25% was found to be optimum for acquiring suitable consistency of the solution, thereby resulting in the formation of nanoparticles of desired size range. Furthermore, different concentrations of surfactants were studied. It was found that the surfactants produced larger particles at low concentrations, while high concentrations led to smaller particle size. Based on these results, emulsification was carried out under the following constant conditions: oil/water ratio 1:10, polymer concentration 0.25%, homogenization speed 25,000 rpm. The molecular weight of the polymer and the type of the emulsifier varied at three different levels. Two different molecular weights of PCL were used: low molecular weight (14,000 g/mol), high molecular weight (80,000 g/mol), and combination of both. The type of the emulsifier also varied in three ways: Poloxamer 407, Polysorbate 20, and a combination of the two. The dependent variables were particle size, size distribution, and ζ-potential.

Nine batches of blank PCL NPs were obtained by single emulsion/solvent evaporation technique to study the influence of the molecular weight and the type of emulsifier. The composition of the obtained PCL NPs is shown in Table 2. It can be observed that, for the stable formulation, optimized polymer and surfactant concentrations are required.

The characteristics of the blank PCL NPs are shown in Table 3 and Figure 2. Mean particle size varied in a range between 188 and 628 nm. Models prepared with a polymer of lower molecular weight have smaller average size, which is probably due to the lower viscosity of the polymer solution used in their preparation. Models (NP1, NP2, NP3) prepared with emulsifier Poloxamer 407 have larger sizes than those (NP4, NP5, NP6) prepared with Polysorbate 20. The effectiveness of the emulsifier depends on its ability to be adsorbed onto the oil/water interface, thereby decreasing the surface tension, and preventing coalescence of newly formed droplets. However, the efficacy also depends on how quickly the emulsifier is adsorbed onto the droplet surfaces. All the droplets are enveloped by a layer of emulsifier when the coarse emulsion first enters the homogenizer, and therefore, the interfacial tension of all the droplets should be relatively low. After the larger droplets are fragmented, the total surface area is greatly increased, the droplets may not be completely covered by emulsifier, thereby leading to a substantial increase in the interfacial tension. If emulsifier molecules in the aqueous phase can be adsorbed onto the droplet surfaces faster than a subsequent fragmentation event, then the interfacial tension will be relatively low and droplet breakdown will be favored [27,28]. However, if the emulsifier molecules are adsorbed too slowly, then the droplet breakdown will be less efficient. Consequently, the ability of the droplets to be disrupted with the help of a homogenizer depends on the adsorption kinetics of the emulsifiers, as well as their ability to decrease the interfacial tension. It is possible that higher molecular weight surfactants move and adsorb slowly onto newly formed droplets; therefore, they are often less effective in reducing particle size during high-speed homogenization. Poloxamer 407 has a molecular weight of 12.6 kDa, whereas Polysorbate 20 molecular weight is 1.22 kDa. In their study, Cirin and Krstonošic noticed that the presence of Poloxamer 407 in polysorbate solutions reduces the concentration necessary to obtain required surface tension value. Such synergism, regarding surface tension reduction efficiency, was attributed to dipole-induced dipole interactions between the hydrophobic moieties of poloxamer and the hydrophobic parts of polysorbate surfactants [29]. A combination of the two emulsifiers led to size reduction, compared to models prepared with Poloxamer 407 alone, but did not result in particles of optimal size. According to the literature, NPs with size between 10 and 300 nm can deliver therapeutic agents through the olfactory region directly to the brain [30]. The mechanisms of nose-to-brain transport of NPs via the olfactory pathway are still unclear. The transport of NPs depends on the NPs’ morphology and surface characteristics [21]. NPs with hydrophilic characteristics are transported paracellularly, while those with hydrophobic properties are transported transcellularly [31].

The polydispersity index (PDI) was measured to determine whether the particles were homogeneously distributed regarding their particle size. A polymodal size distribution was found for the models prepared with Poloxamer 407 as an emulsifier, whereas the models formulated using Polysorbate 20 demonstrated homogenous monomodal particle size distribution (Figure 2).

The ζ potential ranged from −5.3 to −17.9 mV. Negative values of the ζ-potential may originate from the carboxylic terminal group of PCL. Values in the range from −15 mV to −30 mV are considered ideal for stabilization of the nanoparticles. Electrostatic repulsion between the particles is increased with high absolute values of negative ζ-potential, which prevents their aggregation and thus stabilizes the dispersion of nanoparticles. 

Based on the obtained results regarding particle size, PDI, and ζ-potential, the models NP4, NP5, and NP6 (formulated with Polysorbate 20 as an emulsifier) were considered optimal and were selected for further studies.

### 3.2. Synthesis and Characterization of IDB-Loaded PCL NPs

IDB-loaded PCL NPs were prepared via emulsion/solvent evaporation technique. Three batches of drug-loaded NPs were prepared based on the optimized formulations of blank NPs. The polymer/drug ratio was 1:1.25 (Table 4). The results of the study are summarized in Table 5.

#### 3.2.1. Drug Loading and Entrapment Efficiency

Drug loading of the NPs ranged from 21.45% to almost 30%. A tendency of decreasing drug loading was observed with increasing the molecular weight of the polymer, which could be attributed to the higher viscosity of PCL 80 kDa solution compared to that of PCL 14 kDa at the same concentration. It is generally accepted that the mobility of drug molecules slows down in more viscous media, thus preventing entrapment of large amounts of IDB.

Entrapment efficiency of the NPs was mainly dependent on the drug partition coefficient in the internal and external phases of the emulsion [32]. Acceleration of NP solidification may reduce the drug partitioning in the external aqueous phase and increase the percentage of entrapped drug. The entrapment efficiency was strongly dependent on the molecular weight of PCL. Lower entrapment efficiency was observed ranging from 53.22% for PCL 14 kDa NPs (NP4-IDB) to 36.98% for PCL NPs 80 kDa (NP5-IDB). This may be due to the lower compatibility of IDB with PCL of higher molecular weight PCL. Chromatograms of IDB released from nanoparticles of batches NP4-IDB and NP5-IDB are shown in Figure 3. Mixing PCL of different molecular weights (14 kDa and 80 kDa, NP6-IDB) did not substantially increase the drug payload.

#### 3.2.2. FTIR Analysis

The FTIR spectra of IDB, PCL, and IDB-loaded PCL NPs are presented in Figure 4. IDB demonstrated several characteristic peaks at 3568 cm^−1^, 2921 cm^−1^, 2847 cm^−1^, and 1645 cm^−1^ corresponding to O-H, C-H, and C=O groups and stretching vibration at 1611 cm^−1^, a result of the C=C ring. PCL shows two characteristic peaks: one at 2945 cm^−1^, due to C-H stretching vibrations of the methylene groups, and one at 2872 cm^−1^, related to a characteristic single band corresponding to the carbonyl group. In the IDB-loaded PCL NPs, the characteristic peak of PCL at 2945 shifts to 2922 cm^−1^ and that at 2872 cm^−1^ shifts to 2866 cm^−1^ due to an overlap of C-H and C=O groups of IDB. All other characteristic peaks of IDB are present in the nanoparticles, indicating successful incorporation of the drug into the polymer matrix. 

#### 3.2.3. Scanning Electron Microscopy and Transmission Electron Microscopy

SEM was performed to verify DLS particle size data and investigate nanoparticles’ geometry. The SEM micrograph showed the spherical shape of the nanoparticles and uniform size distribution (Figure 5). TEM images further confirmed the production of spherical nanoparticles (Figure 6).

#### 3.2.4. In Vitro Drug Release

The dissolution profiles of IDB from PCL NPs are presented in Figure 7. The percentage of released drug during the 48 h study was incomplete, varying from 50% (NP5-IDB) to 60.07% (NP4-IDB). From the graph, it can be observed that there is a gradual increase in the amount of IDB released from the NPs, which may be due to the cleavage of ester linkages of PCL. No burst effect was observed, which, given the intended route of administration through the olfactory region of the nasal cavity directly to the brain, allows the drug to remain unreleased at the site of administration and suggests its release primarily in the targeted tissue [21]. However, further studies involving animal experimental models are needed to prove this hypothesis.

The in vitro release study indicated that the drug release rate decreased upon increasing the molecular weight of the polymer. The release rate from the NPs prepared from high molecular weight PCL (NP5-IDB) was slow relative to that of the samples prepared from low molecular weight PCL (NP4-IDB). These findings could be attributed to the higher viscosity of the polymer layer formed around the drug upon contact with the dissolution medium. It should be considered that the release of drug molecules from polyesters such as PCL can be incomplete because of its hydrophobicity and higher crystallinity. Polymer–drug interactions, drug solubility in the medium, and polymer interactions with the medium must also be considered to elucidate the drug release kinetics. 

The mechanism of IDB release from the PCL NPs was investigated by fitting the in vitro drug release data into different kinetic models (zero order, first order, Higuchi, Korsmeyer–Peppas, and Hixson–Crowell). The R-squared values calculated for the different kinetic models are presented in Table 6. Data analysis revealed that IDB release from the NPs followed Korsmeyer–Peppas release kinetics, which is typical for polymer-based drug carriers. The values of the release exponent (*n*) were calculated (*n* = 0.4817 for model NP4-IDB and *n* = 0.5288 for model NP5-IDB). The value *n* may provide information about the physical mechanism controlling the drug release from the NPs. Based on the value of this exponent, drug release was controlled by non-Fickian diffusion (diffusion coupled with erosion).

## 4. Conclusions

In the present study, IDB-loaded PCL NPs were successfully prepared by single emulsion/solvent evaporation technique with high entrapment efficiency and optimal particle size for nose-to-brain delivery. The optimized IDB-loaded PCL NPs demonstrated prolonged drug release and followed non-Fickian diffusion-based kinetics. Based on these findings, the formulated nanoparticles proved their potential for nose-to-brain delivery of IDB. Further studies are ongoing to develop a suitable nanocomposite drug delivery platform that meets the requirements of nasal formulation in terms of deposition site and mucoadhesive property.

## Figures and Tables

**Figure 1 biomedicines-11-01491-f001:**
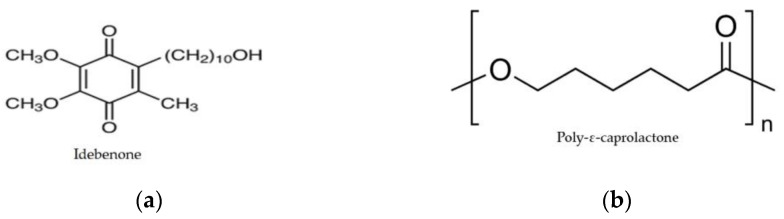
Chemical structures of idebenone (**a**) and poly-ε-caprolactone (**b**).

**Figure 2 biomedicines-11-01491-f002:**
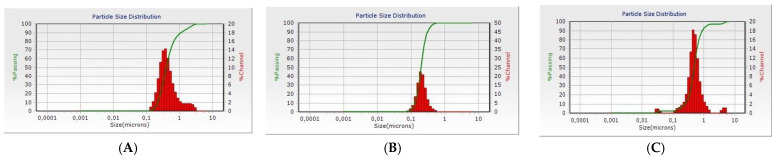
DLS histograms of blank PCL NPs of batches NP1 (**A**), NP4 (**B**), NP7 (**C**).

**Figure 3 biomedicines-11-01491-f003:**
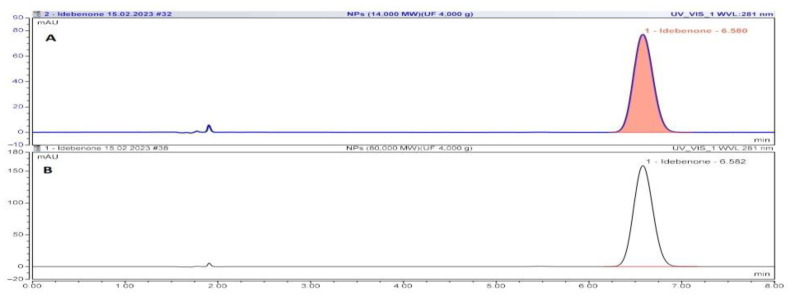
Chromatograms of IDB released from nanoparticles of models NP4-IDB (**A**) and NP5-IDB (**B**).

**Figure 4 biomedicines-11-01491-f004:**
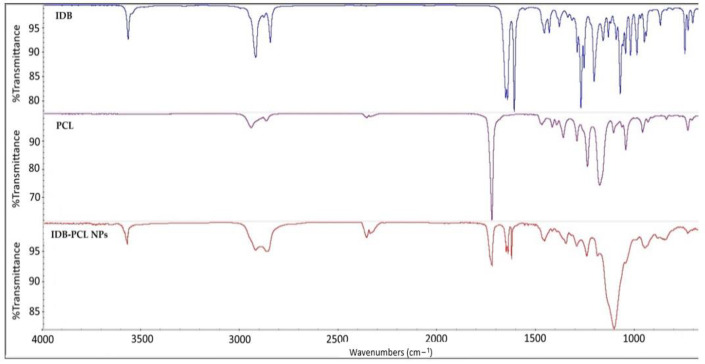
FTIR spectra of IDB, PCL, and IDB-PCL NPs.

**Figure 5 biomedicines-11-01491-f005:**
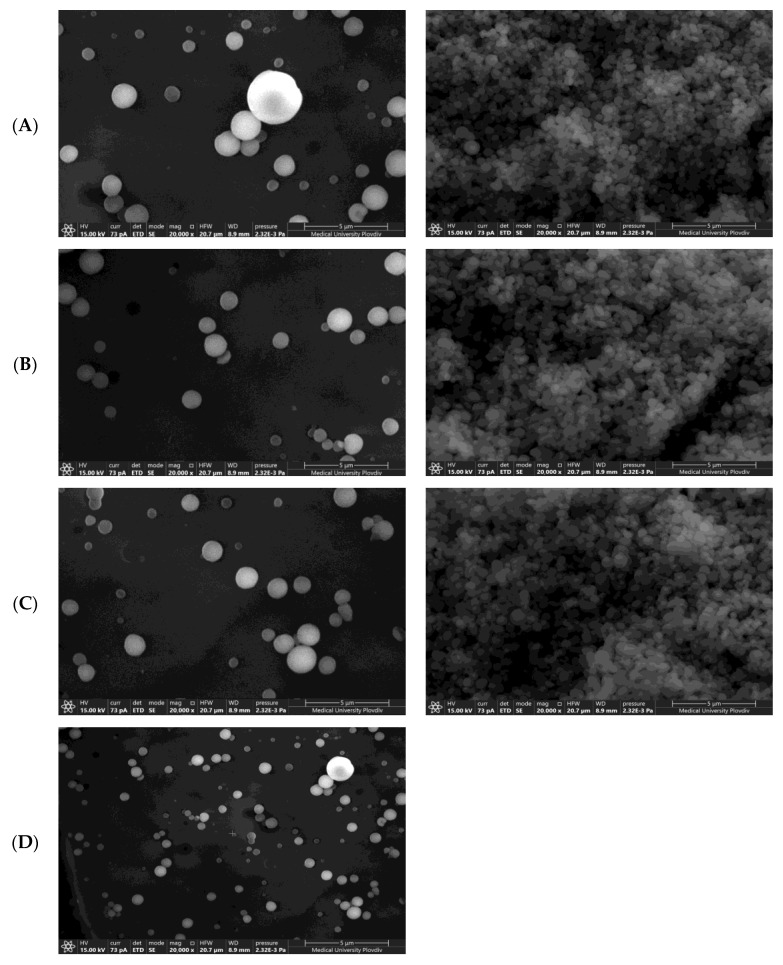
SEM micrographs of blank PCL NPs in suspension (**left**) and dried NPs (**right**) of batches NP1 (**A**), NP4 (**B**), NP7 (**C**), and IDB-loaded PCL NPs (**D**), magnification 20,000×.

**Figure 6 biomedicines-11-01491-f006:**
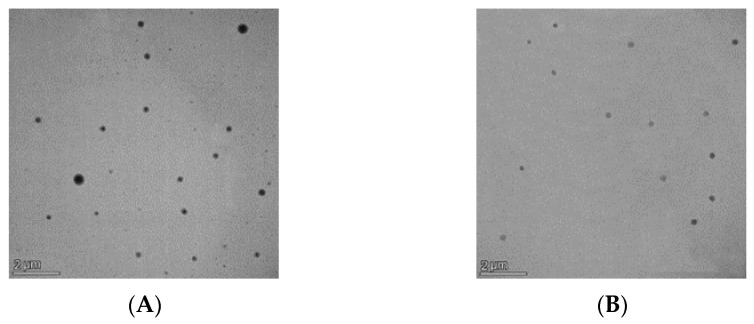
TEM micrographs of blank (**A**) and IDB-loaded (**B**) PCL NPs, magnification 20,000×.

**Figure 7 biomedicines-11-01491-f007:**
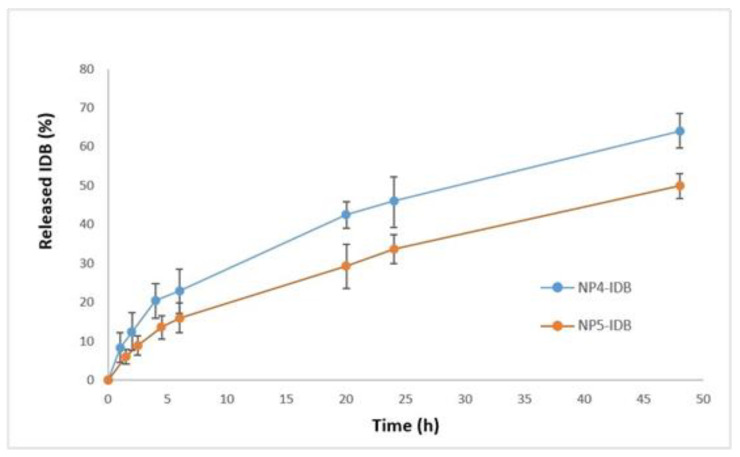
In vitro IDB release from NPs prepared from PCL with different molecular weights.

**Table 1 biomedicines-11-01491-t001:** Preliminary experimental work to derive optimal conditions.

Conditions	Results
**Oil/water ratio**	
1:2	Unstable emulsion
1:5	Stable emulsion, larger particles
1:10	Stable emulsion, optimal particle size
**Stirring speed**	
16,000 rpm	Larger particles
25,000 rpm	Optimal particle size
35,000 rpm	High dispersity, very small particle size
**PCL concentration**	
0.10%	Larger particles
0.25%	Optimal particle size
0.50%	Larger particles
**Poloxamer 407 concentration**	
2.0%	Unstable emulsion
4.0%	Stable emulsion
6.0%	Stable emulsion
**Polysorbate 20 concentration**	
0.5%	Stable emulsion
1.0%	Stable emulsion
1.5%	Stable emulsion

**Table 2 biomedicines-11-01491-t002:** Composition of blank PCL NPs.

Composition (%)	Sample Code								
	NP1	NP2	NP3	NP4	NP5	NP6	NP7	NP8	NP9
PCL (14 kDa)	0.25	-	0.125	0.25	-	0.125	0.25	-	0.125
PCL (80 kDa)	-	0.25	0.125	-	0.25	0.125	-	0.25	0.125
Poloxamer 407	4.00	4.00	4.000	-	-	-	2.00	2.00	2.000
Polysorbate 20	-	-	-	1.00	1.00	1.000	0.20	0.20	0.200

**Table 3 biomedicines-11-01491-t003:** Characteristics of blank PCL NPs (mean values ± SD, *n* = 3; PDI (polydispersity index)).

Sample Code	Mean Diameter (nm)	PDI	ζ-Potential (mV)
NP1	474 ± 250	17.73	−5.3 ± 0.2
NP2	628 ± 380	22.30	−7.9 ± 0.9
NP3	536 ± 204	5.75	−10.9 ± 0.3
NP4	188 ± 61	0.84	−14.5 ± 0.5
NP5	201 ± 80	1.04	−16.9 ± 0.2
NP6	196 ± 70	0.91	−17.9 ± 0.7
NP7	336 ± 150	2.43	−6.7 ± 0.8
NP8	398 ± 250	3.72	−8.5 ± 0.5
NP9	390 ± 200	3.05	−9.5 ± 0.2

**Table 4 biomedicines-11-01491-t004:** Composition of drug-loaded NPs.

IDB (mg)	IDB (%)	PCL (mg)	PCL (%)	IDB:PCL
10	0.2	12.5	0.25	1:1.25

**Table 5 biomedicines-11-01491-t005:** Characteristics of IDB-loaded PCL NPs (mean values ± SD, *n* = 3); PDI—polydispersity index, EE—entrapment efficiency, DL—drug loading.

Sample Code	Mean Diameter (nm)	PDI	ζ-Potential (mV)	EE (%)	DL (%)
NP4-IDB	195 ± 80	0.82	−13.7 ± 0.3	53.22 ± 0.60	29.86 ± 0.20
NP5-IDB	212 ± 90	1.11	−15.3 ± 0.9	36.98 ± 0.57	22.83 ± 0.22
NP6-IDB	205 ± 60	1.25	−17.4 ± 0.7	34.15 ± 0.23	21.45 ± 0.31

**Table 6 biomedicines-11-01491-t006:** R^2^ values evaluated for kinetic modelling of in vitro drug release studies.

Sample Code	Zero Order	First Order	Higuchi	Korsmeyer–Peppas	Hixson–Crowell
NP4-IDB	0.9490	0.9872	0.9985	0.9973	0.9774
NP5-IDB	0.9683	0.9912	0.9986	0.9992	0.9853

## Data Availability

Not applicable.

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
