# Peer review of "Synthesis and Characterization of PCL-Idebenone Nanoparticles for Potential Nose-to-Brain Delivery"

_biomedicines, 2023, doi:10.3390/biomedicines11051491_

Round 1
Reviewer 1 Report
The manuscript by Boyuklieva et al. discusses the preparation of an optimal model of poly-ε-caprolactone (PCL) nanoparticles as potential carriers for nasal administration of idebenone. Three models of idebenone-loaded nanoparticles were developed and the effect of molecular weight on the encapsulation efficiency was investigated. Increased encapsulation efficiency was found when PCL with lower molecular weight was used. Dissolution study data was fitted into various kinetic models and the Korsmeyer-Peppas model was found to be the indicative of the release mechanism of idebenone. The claims are mostly supported, and the conclusions appear to be sound. My main suggestion for improvement is the quality of data presentation. Overall, I suggest minor revision, and here’s a list of questions and improvements that the authors could consider:
- The mean particle size has quite a big range of 188~628 nm, how consistent is this result, and is there any way or mechanistic benefit to improve this monidispersity?
- The authors are encouraged to include more SEM and TEM images, as the current ones only represent a small portion of the result, a statistical analysis is preferred.
Author Response
The authors are grateful for the reviewer's valuable comments. The responses to the reviewer commnts are provided below:
Q1: The mean particle size has quite a big range of 188~628 nm, how consistent is this result, and is there any way or mechanistic benefit to improve this monidispersity?
R1: We fully agree with the reviewer's remark that the particle size ranges widely, and we have thoroughly investigated the impact of different variables. As explained in the discussion section of the article, the particles’ size was affected by:
-the molecular weight (Mw) of the polymer- as the Mw increases, the particle size increases, probably due to the higher viscosity of the polymer solution used for preparation;
-the type of the emulsifier - batches prepared with Poloxamer 407 have larger sizes than those prepared with Polysorbate 20. The effectiveness of the emulsifier depends on its ability to be adsorbed on the oil/water interface, thereby decreasing the surface tension, and preventing coalescence of newly formed droplets. However, the efficacy also depends on how quickly the emulsifier is adsorbed onto the droplet surfaces. All the droplets are enveloped by a layer of emulsifier when the coarse emulsion first enters the homogenizer, and therefore the interfacial tension of all the droplets should be relatively low. After the larger droplets are fragmented, the total surface area is greatly increased, the droplets may not be completely covered by emulsifier, thereby leading to a substantial increase in the interfacial tension. If emulsifier molecules in the aqueous phase can be adsorbed onto the droplet surfaces faster than a subsequent fragmentation event, then the interfacial tension will be relatively low and droplet breakdown will be favored. However, if the emulsifier molecules are adsorbed too slowly, then the droplet breakdown will be less efficient. Consequently, the ability of the droplets to be disrupted with the help of a homogenizer depends on the adsorption kinetics of the emulsifiers, as well as their ability to decrease the interfacial tension. It is possible that higher molecular weight surfactants move and adsorb slowly to newly formed droplets, therefore they are often less effective in reducing particle size during high-speed homogenization. Poloxamer 407 has a molecular weight of 12.6 kDa, whereas Polysorbate 20 molecular weight is 1.22 kDa.
Based on these findings, we concluded, that the models prepared with polysorbate 20 as an emulsifierdemonstrated optimal physicochemical properties for nose-to-brain delivery, size in the range of 188-201 nm, and therefore they were loaded with Idebenone.
Q2: The authors are encouraged to include more SEM and TEM images, as the current ones only represent a small portion of the result, a statistical analysis is preferred.
R2: SEM micrographs from different batches in suspension and in a dry state were added.
Reviewer 2 Report
Nanomedicine has become increasingly popular in recent years. Poly(ε-caprolactone) is a convenient and widely used basis for obtaining nanomaterials. Expanding the possibilities of its use is an urgent task (Poly(ε-caprolactone) (PCL) Hollow Nanoparticles with Surface Sealability and On-Demand Pore Generability for Easy Loading and NIR Light-Triggered Release of Drug, Pharmaceutics 2019, 11(10), 528; https://doi.org/10.3390/pharmaceutics11100528, Design, development, and characterization of an idebenone-loaded poly-ε-caprolactone intravitreal implant as a new therapeutic approach for LHON treatment, European Journal of Pharmaceutics and Biopharmaceutics 168 (2021) 195–207, https://doi.org/10.1016/j.ejpb.2021.09.001, Poly ε-Caprolactone Nanoparticles for Sustained Intra-Articular Immune Modulation in Adjuvant-Induced Arthritis Rodent Model, Pharmaceutics. 2022 Mar; 14(3): 519. https://doi.org/10.3390/pharmaceutics14030519, Poly(ε-caprolactone) nanocapsule carriers with sustained drug release: single dose for long-term glaucoma treatment, Nanoscale. 2017, https://doi.org/10.1039/C7NR03221H).
This manuscript is devoted to the development of a new dosage form of a drug known in pharmacology. The description of the methods is given in detail and allows you to reproduce the key points to other performers. Optimization of the method for obtaining nanomaterials was also carried out, the most effective composites for the inclusion of the drug and its stability of the nanocomposite were selected.
The obtained nanomaterials are well and sufficiently characterized. The entry of the target drug into the biological environment from the synthesized nanomaterials has also been studied.
The manuscript is well written and will be of interest to a wide range of specialists and can be accepted for publication.
Author Response
The authors are grateful to the reviewer for his/her positive opinion on the research.